# 'Survival-driven' or 'policy-driven'—research on the motivation and economic consequences of the mixed-ownership reform of state-owned enterprises

Xiuting Wang *

Department of Accounting, Shandong Technology and Business University, Yantai, Shandong, China

* wangxiuting@sdtbu.edu.cn

## Abstract

Mixed-ownership reform (MOR) represents a pivotal policy instrument for revitalizing China's state-owned enterprises (SOEs); however, its outcomes exhibit substantial heterogeneity. This variation implies that the intrinsic motivations behind reform—whether prompted by regulatory compliance or economic necessity—may constitute an understudied determinant of its efficacy. This research empirically examines how reform motivations influence the economic outcomes of mixed-ownership restructuring. Utilizing panel data from Chinese listed SOEs between 2013 and 2022 and employing a double fixed-effects model, we analyze the interaction between reform attributes—namely, breadth (ownership diversity) and depth (proportion of nonstate ownership)—and reform motivations, classified according to prereform profitability. Our results indicate that in SOEs characterized by survival-driven motives, deeper reform significantly improves firm performance. Nonetheless, this positive relationship is moderated by regional marketization levels. Conversely, reforms motivated primarily by policy directives demonstrate weaker effects on performance. These findings underscore that the impacts of mixed-ownership reform are not monolithic but are fundamentally conditioned on initial motivations and external institutional contexts. This study contributes to a refined theoretical framework for interpreting the heterogeneous effects of SOE reform and offers actionable implications for formulating context-sensitive policies.

## 1. Introduction

As a pivotal initiative in China's state-owned enterprise (SOE) reforms, mixed-ownership reform (MOR) has attracted significant attention from policymakers, academics, and industry practitioners. The literature has extensively examined the economic consequences of mixed reform, particularly its effects on corporate governance and firm performance [1–3]. Building upon these foundations, this study seeks

**Data availability statement:** All relevant data are within the manuscript and its Supporting Information files.

**Funding:** Humanities and Social Sciences Project of Shandong Province: Research on the Mixed Ownership Reform of State owned Enterprises in Shandong Province: Motivation, Effect, and Improvement Strategy (2022-YYGL-37).

**Competing interests:** The authors have declared that no competing interests exist.

to extend the current understanding by exploring the underlying motivations that drive SOEs to pursue ownership restructuring and how these motivations shape reform outcomes.

Prior research has demonstrated that mixed ownership can improve firm performance through enhanced monitoring [4,5], resource integration [6], and innovation capacity [7,8]. However, these studies have predominantly treated mixed reform as an exogenous policy intervention, paying limited attention to the endogenous motivations that influence SOEs' reform decisions. This perspective may help explain the mixed empirical evidence regarding reform effectiveness [9–11], suggesting the need for a more nuanced examination of reform implementation.

Drawing on institutional theory [12] and resource-dependence theory [13], this study proposes a conceptual framework that distinguishes between policy-driven reforms (where SOEs respond to external governmental pressure) and survival-driven reforms (where SOEs proactively initiate changes to address operational challenges). We suggest that these motivations interact differently with internal governance mechanisms and external market conditions, leading to varying reform pathways and outcomes.

Our empirical investigation employs a dual fixed-effects model to analyze panel data from Chinese SOEs between 2013 and 2022. Following established methodologies [14,15], we measure reform breadth through shareholder diversity and reform depth through nonstate ownership concentration while controlling for firm-level characteristics and temporal trends.

This research aims to contribute to the literature in several ways. First, by integrating institutional and resource dependence perspectives, we offer a more comprehensive theoretical framework for understanding SOE reform behavior. Second, our findings may provide practical insights for policymakers seeking to design more effective and contextually appropriate reform strategies, particularly regarding the sequencing and implementation of MORs in different regional contexts [16]. Third, the study highlights the importance of aligning reform measures with local institutional environments, suggesting that ownership restructuring can complement market institutional development in promoting SOE efficiency [17].

The remainder of this paper proceeds as follows. Section 2 develops our theoretical framework and hypotheses. Section 3 describes our research design and empirical methodology. Section 4 presents our main results and robustness tests. Section 5 discusses additional analyses examining mediating mechanisms and boundary conditions. Section 6 concludes with theoretical and policy implications.

## 2. Theoretical analysis and research hypotheses

MOR encompasses two distinct transformation paths: quantitative change (breadth), which represents the diversification of shareholder composition, and qualitative change (depth), which implies the deepening of the governance structure through mutual checks and balances among heterogeneous shareholders [6]. The efficacy of these paths, however, is not automatic but is theoretically contingent on the underlying motivations driving the reform. Drawing on institutional theory and

resource-dependence theory, this section develops a nuanced framework to hypothesize how reform motivations moderate the relationship between reform characteristics (breadth/depth) and their outcomes.

## 2.1. MOR breadth and its effects

MOR breadth, defined as the variety of equity types introduced (e.g., state, private, institutional, or foreign), aims to inject diversity into the corporate governance structure. The capitalistic nature of nonstate shareholders drives them to seek maximum investment returns [9]. This pursuit aligns with the core tenets of agency theory [18,19], which posits that conflicts of interest arise between principals (owners) and agents (managers). In SOEs, the owner absence problem and entrenched insider control often lead to managerial slack and inefficient investments [7,20]. The introduction of diverse, profit-oriented nonstate shareholders can mitigate these agency costs by enhancing monitoring intensity and imposing market discipline on managerial decisions [21,22]. For instance, nonstate shareholders are motivated to scrutinize and optimize investment decisions, production processes, and financial management to safeguard their property rights and improve returns [9,23].

Furthermore, resource-dependence theory [24] suggests that organizations rely on external resources for survival and success. A broader equity structure helps SOEs reduce their dependence on a single resource provider (the state) and access critical resources possessed by new shareholders, such as market knowledge, technology, and management expertise [25,26]. This resource infusion is crucial for overcoming the inefficiencies stemming from SOEs' social burdens and nonmarket-oriented operational mechanisms [10,11].

From the perspective of human capital theory [27], the breadth of reform also promotes the construction of market-oriented incentive mechanisms. Nonstate shareholders, accustomed to performance-sensitive compensation in private and foreign firms [12], advocate linking executive pay to financial performance. This helps overcome the weak pay-performance sensitivity in SOEs, where managerial incentives are often tied to political promotion rather than economic performance [13,28]. By fostering a talent selection and evaluation system that rewards merit, a broader shareholder base enhances managerial motivation and operational efficiency.

Therefore, we propose that a wider MOR breadth introduces stronger monitoring, diversifies critical resources, and aligns managerial incentives with market performance.

**H1: There is a positive correlation between MOR breadth and its effectiveness in improving SOE performance.**

## 2.2. MOR depth and its effects

MOR depth, operationalized as the proportion of nonstate-owned capital infusion, is posited to be a critical determinant of reform effectiveness. This relationship is not merely quantitative but is underpinned by a robust theoretical framework encompassing corporate governance, economics, and strategic management. A deeper level of equity integration signifies a substantive transfer of influence to nonstate shareholders, which is a prerequisite for achieving meaningful governance transformation and performance enhancement [14,15].

The theoretical justification for the impact of reform depth can be articulated through three primary mechanisms:

**2.2.1. Governance empowerment and stewardship effects.** A superficial level of nonstate ownership may grant limited voice, but substantial shareholding is necessary to empower these shareholders to effectively address inherent issues within SOEs, such as owner vacancy and insider control [14,20]. This aligns with the principles of stewardship theory [29], which suggests that when stakeholders possess significant ownership and influence, they are more likely to act as stewards of the enterprise, contributing proactively to long-term value creation rather than engaging in short-term opportunism [30]. Furthermore, from an agency theory perspective [18], a higher ownership stake increases the incentives and capabilities of nonstate shareholders to monitor management rigorously, thereby reducing agency costs and

mitigating the risk of managerial expropriation [21,31]. Only with sufficient depth can nonstate capital transition from being passive investors to being active governance actors, enabling genuine mutual supervision and checks and balances.

### 2.2.2. Resource complementarity and knowledge spillovers.
From an economic standpoint, a deeper reform facilitates a more substantial influx of the strategic resources that nonstate capital possesses. This is a core tenet of the resource-based view (RBV) and knowledge spillover theory [32,33]. Nonstate investors often bring not only capital but also advanced technologies, market expertise, and efficient management practices. A critical mass of ownership is often required to ensure the transfer and deep integration of these valuable resources into the SOE's operations [25,26]. For instance, the case of China Unicom's profound MOR, which involved significant private and strategic investor participation, led to notable improvements in market competitiveness and innovation, illustrating the resource complementarity achieved through deep integration [1].

### 2.2.3. Reduction in transaction costs and enhanced decision-making quality.
A balanced and substantial equity structure among heterogeneous shareholders can lead to more efficient governance. Drawing on transaction cost economics (TCE) [34], deeper integration reduces the costs associated with negotiation, coordination, and monitoring between state and nonstate shareholders. A more balanced power distribution fosters more scientific, democratic, and transparent decision-making processes, curbing the arbitrary power of insiders and reducing the likelihood of suboptimal investments [6,35]. This creates a more stable and predictable governance environment that is conducive to sustainable growth.

In summary, the depth of the MOR is theorized to empower nonstate shareholders, facilitate critical resource integration, and create a more efficient governance system. Therefore, we formally propose the following hypothesis:

**H2: There is a positive correlation between MOR depth in SOEs and the effectiveness of the reform. Specifically, a greater proportion of nonstate-owned capital (i.e., greater depth) leads to a more significant improvement in SOE performance.**

### 2.3. Motivation, degree, and effects of MOR

The ultimate impact of the breadth and depth of MOR is not uniform but is critically shaped by the primary motivation behind the reform. We classify motivations into two archetypes on the basis of institutional theory [36] and resource-dependence theory [24], which provide a robust framework for understanding the divergent paths and outcomes of reform.

Policy-Driven (Passive) MOR: A Quest for Legitimacy. This motivation arises primarily from top-down institutional pressures to comply with national reform agendas [17,37–41]. SOEs undertaking reform for this reason often respond to coercive isomorphism [36], a concept from institutional theory in which organizations change their formal structures to gain legitimacy and conform to external expectations rather than improving internal efficiency. In such cases, the reform may be largely ceremonial. Nonstate shareholders might be introduced to meet quantitative targets ("breadth") but are granted limited real influence ("depth") [42,43], a phenomenon consistent with decoupling [44], where policy mandates are symbolically adopted without substantive changes in core activities. Furthermore, the participation of nonstate shareholders itself might be motivated by a desire to access the SOE's political resources and social reputation (resource dependence) rather than fundamentally improving governance [4]. Consequently, the potential governance and resource benefits predicted by agency and resource dependence theories are likely to be diluted. This reform lacks the internal drive for genuine transformation, as the primary goal is legitimacy acquisition rather than operational enhancement [45,46].

Survival-Driven (Active) MORs: A Response to Resource Scarcity. This motivation stems from acute internal crises, such as poor profitability, illiquidity, and inefficient management systems [47–53]. From a resource-dependence theory perspective, these crises represent critical resource constraints and threats to organizational survival [24]. Faced with such threats, SOEs undertake reform out of a pressing need to acquire essential external resources, managerial capabilities, and market discipline that they lack internally [25,26]. Here, the introduction of nonstate capital is a strategic,

necessity-driven response. Management and nonstate shareholders have a strong, aligned interest in implementing substantive reforms in terms of governance, incentives, and innovation to ensure enterprise survival and turnaround [54]. This alignment creates a conducive environment for the mechanisms of agency reduction and resource complementarity to function effectively. The reform is characterized by a genuine need for the depth of influence granted to new shareholders and the breadth of the resources they bring, ensuring that structural changes translate into performance improvements [55,56].

The core theoretical argument is that survival-driven motivation, rooted in resource necessity, creates internal conditions for substantive change, whereas policy-driven motivation, rooted in legitimacy seeking, often leads to symbolic compliance. This differential implementation of reform structures explains why the same reform characteristics (breadth and depth) yield stronger effects under survival-driven conditions. The mechanism flowchart illustrating the outcomes of MOR is shown in Fig 1.

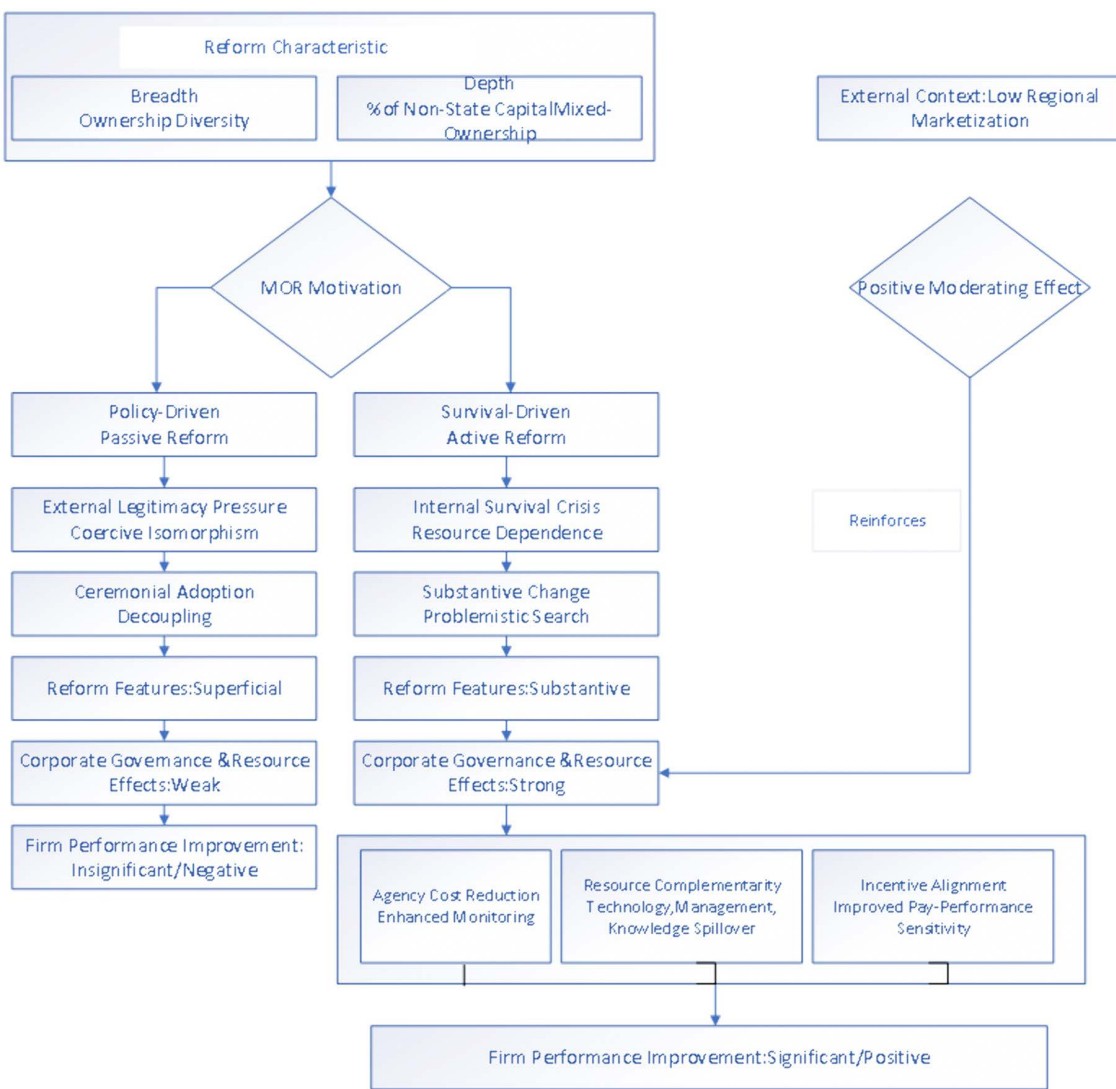

**Fig 1. Mechanism flowchart of mixed-ownership reform (MOR) effects.**

This leads to our final set of hypotheses concerning the moderating role of motivation:

**H3a: The positive relationship between MOR breadth and reform effectiveness is stronger for survival-driven SOEs than for policy-driven SOEs.**

**H3b: The positive relationship between MOR depth and reform effectiveness is stronger for survival-driven SOEs than for policy-driven SOEs.**

## 3. Research design

### 3.1. Sample selection and data source

This study uses the Decision of the Central Committee of the Communist Party of China on Several Major Issues Concerning Comprehensively Deepening Reform, which was issued during the Third Plenary Session of the 18th Central Committee of the Communist Party of China, as a catalyst. This decision initiated the process of MOR in SOEs, on the basis of which a quasinatural experimental model is constructed. State-owned listed enterprises from 2013 to 2023 are selected as the research sample to analyze the motivations, degree, and effects of SOE MOR. The sample selection process is as follows: First, the list of SOE MOR pilot enterprises identified by the National Development and Reform Commission (NDRC) and the State-owned Assets Supervision and Administration Commission (SASAC) is manually collected. Afterward, the reform list is supplemented, and indicators such as MOR breadth and MOR depth are calculated via detailed data from the CSMAR database, including listed company announcements, corporate governance information, and equity structure data. Finally, information is manually collected through channels such as news reports, corporate announcements, and media disclosures, including whether a listed company participated in MOR, the year of participation, the initial announcement date of the reform, the progress of reform, and the list of introduced investors. The statistical software used included Jupyter Notebook in Anaconda3, Stata 15, and Excel 2019.

Following the practices of previous studies, the raw data are processed as follows:

(1) Exclude samples from the financial industry;

(2) Remove observations of enterprises that were classified as ST or *ST during the sample period;

(3) Delete samples with missing values for key variables;

(4) Apply winsorization to all continuous variables at the 1% and 99% levels.

Ultimately, 4,124 firm-year observations are obtained.

### 3.2. Variable definitions

**3.2.1. Dependent variable: MOR effect.** The effect of MOR is measured primarily through market value and accounting profits. First, the **market value** is captured by **Tobin's Q**, which is derived from the CSMAR database and defined as the ratio of market value to total assets (i.e., market value/total assets). Second, **accounting profit indicators** include return on assets (ROA), return on equity (ROE), and operating profit margin (OPR). While both ROA and ROE are important measures of corporate performance, ROE considers only shareholders' equity in the denominator, whereas ROA better reflects the efficiency of capital utilization. Drawing on Zhu Jingyi (2024) [18], ROA is selected as the indicator to evaluate the effect of the reform. In the benchmark model, the MOR effect is measured using ROA, whereas Tobin's Q is used for robustness tests.

**3.2.2. Independent variables. ① MOR breadth (breadth).** Following the approach of Cao Yue et al. (2020), MOR breadth is defined as the number of nonpublic ownership equity types in SOEs, quantifying the "coverage" of the reform (i.e., the degree of "mixing"). The equity structure is subdivided into four categories. The first is state-owned equity (SOE),

which includes equity directly controlled by the state through government agencies such as the State-owned Assets Supervision and Administration Commission (SASAC), as well as equity indirectly held by state-led entities or wholly owned government asset management companies. The second is individual/family equity (person/family), which refers to equity held by domestic natural persons or families through investment. The third is foreign equity (foreigner), defined as equity held by overseas-registered legal entities (including those registered in Hong Kong, Macao, and Taiwan, as well as foreign legal entities) and foreign natural persons in mainland listed companies in accordance with relevant regulations. Finally, nonstate-owned enterprise equity (non-SOE) involves equity formed by investments from mainland nonpublic legal entities (e.g., nongovernmental organizations, public institutions, and financial institutions).

The variable extent represents the number of shareholder categories with different ownership natures in reformed SOEs, taking values of {1, 2, 3, 4} [37]. Generally, a more diverse equity composition indicates a greater MOR breadth. For annual reports of listed SOEs in the Shanghai and Shenzhen A-share markets, shareholder information (especially the top 10 shareholders) is typically classified into the categories "state shares," "state-owned legal person shares," "domestic legal person shares," "negotiable A-shares," "foreign shares," "others," and "unknown." We systematically collect and organize the equity attributes and shareholdings of the top 10 shareholders of listed SOEs using online platforms such as Baidu, official corporate websites, Tianyancha, and Qichacha. Where the top 10 shareholders include listed companies, we further analyze the nature of their major controlling shareholders.

② **MOR depth (depth).** MOR depth reflects the level of equity integration and checks-and-balances in SOE reforms. Drawing on Hao Yang et al. (2017), two methods are adopted: First, the aggregate shareholding ratio of nonstate-owned shareholders among the top 10 shareholders before the reform (depth) is taken, where a higher value indicates a greater degree of integration between state capital and nonstate capital, as well as stronger checks exerted by nonstate-owned stakeholders on state-owned shareholders. Second, the Herfindahl–Hirschman index (HHI) for equity distribution in SOEs is calculated as shown in Formula (1). A higher HHI value reflects a higher level of equity integration and checks and balances among different ownership types, i.e., deeper cooperation among shareholders of diverse natures.

In the baseline model, MOR depth is measured using the equity allocation Herfindahl index (MixHHI). To ensure analytical reliability, the aggregate shareholding ratio of nonstate-owned shareholders among the top 10 shareholders of reformed enterprises is used as an alternative indicator for validation.

$$HHI = 1 - \sum p_i^2$$

(1)

where $p_i$ represents the ratio of the shareholding percentage of the i-th type of shareholder among the top ten major shareholders of the state-owned enterprise to the total shareholding percentage of these top ten major shareholders.

③ **Motivations for MOR (motive).** Following the theoretical delineation of policy-driven and survival-driven reforms, their empirical examination necessitates observable and quantifiable measures. We posit that prereform financial performance serves as a robust, externally valid proxy for distinguishing these motivational archetypes at the population level. Accordingly, we operationalize reform motivation dichotomously: SOEs reporting a negative net profit in the fiscal year preceding the reform are classified as survival-driven (coded as 1), whereas those with a positive net profit are classified as policy-driven (coded as 0). This operationalization enables the partitioning of the full sample into distinct cohorts for subsequent contingent analysis.

The rationale for this measurement approach is underpinned by a confluence of theoretical and methodological considerations. First, severe and persistent financial distress constitutes the most salient and unambiguous indicator of an organizational survival crisis. This condition precipitates fundamental problemistic search, rendering MOR a strategic imperative for alleviating immediate existential threats; consequently, the reform motivation in such contexts is inherently potent and endogenous. Despite the possibility that a subset of profitable SOEs may pursue reform proactively, the primary impetus for reform in the absence of financial distress is statistically more likely to be external policy pressure or

the pursuit of ancillary benefits, such as enhanced political capital for executives. Thus, while not exhaustive, this dichotomous classification effectively captures the predominant motivational tendency within each cohort, providing a statistically powerful basis for comparative analysis. Furthermore, the use of prereform profitability as a discriminant variable is consistent with established methodological practices in corporate governance and strategic management research [e.g., following the approach of Dai Fei, 2018]. It offers a parsimonious yet theoretically grounded measure that facilitates large-sample empirical testing while mitigating concerns related to subjective measurement.

**3.3.3. Control variables.** The operational status of an enterprise can directly or indirectly affect corporate value. To mitigate the impact of these factors on the model, drawing on the research of Yang Xingquan et al. (2018), this study controls for the potential influences of factors such as growth (Grow), firm size (Size), asset–liability ratio (Lev), board size (Board), and listing age (Age) on cash holdings, with specific definitions presented in Table 1.

### 3.3. Model design

To examine the relationship between the MOR breadth and MOR depth of SOEs and reform effects under different MOR motivations, the following fixed-effects model is constructed in this study:

$$tobinQ_{i,t} = \alpha_0 + \alpha_1 extent_{i,t} + \alpha_2 HHI_{i,t} + \alpha_3 motive_{i,t}$$
$$+ \alpha_4 motive_{i,t} * extent_{i,t} + \alpha_5 motive_{i,t} * HHI_{i,t} + \alpha_6 \sum controls_{i,t}$$
$$+ \sum YEAR + \sum IND + \varepsilon_{i,t}$$

$$(2)$$

Indices i and t represent the enterprise and year, respectively. To prevent the influence of potential omitted factors, this study implemented fixed-effects adjustment of time and industry and conducted regression analysis through a fixed-effects model of panel data. Controls represents a set of control variables that have an impact on the effectiveness of MOR, with YEAR and IND corresponding to fixed effects of year and industry and ε representing the random error term.

**Table 1. Variable definition table.**

| Variable type | Variable Symbol | Variable | Variable Definition |
|---|---|---|---|
| Dependent Variable | tobinQ | Tobin Q | Market Value/Total Assets |
| | ROA | Return on Assets | Net Profit/Average Total Assets |
| Independent Variable | extent | Breadth of Mixed-ownership Reform | The number of types of shareholders with different natures, whose values may be {1, 2, 3, 4} |
| | depth | Mixed-modification Depth 1 | The sum of the shareholding ratios of non-state-owned shareholders among the top ten shareholders of mixed-ownership SOEs. |
| | HHI | Mixed-modification Depth 2 | Herfindahl index of state-owned enterprise equity structure |
| | motive | Mixed-ownership Motivation | 1 representing survival needs (proactive mixed ownership), 0 representing policy promotion (passive mixed-ownership reform) |
| Control Variable | Grow | Growth | (Current Year's Operating Income – Previous Year's Operating Income)/Previous Year's Operating Income |
| | Size | Company Size | Natural logarithm of total assets at the end of the period |
| | Lev | Financial Leverage Coefficient | Total liabilities at the end of the period/Total assets at the end of the period |
| | Age | Enterprise Listing Age | Add 1 to the difference between the observation year and the company's listing year and take the natural logarithm |
| | Board | Board Size | Add 1 to the number of board members and take the natural logarithm |

# 4. Empirical result analysis

## 4.1. Descriptive statistics

As shown in Table 2, the mean ROA of the sample enterprises is 3.11%, with a minimum value of −63.5% and a maximum value of 58.5%, indicating significant differences in the MOR effect among the sample SOEs. Similarly, the minimum value of the Tobin's Q value of the sample enterprises is 0.625, and the maximum value is 26.82, once again confirming the conclusion of significant differences in the MOR effect. Therefore, studying the MOR effect on SOEs has important practical significance. The mean extent of the MOR is 2.987, indicating that, on average, there are 2 or 3 types of shareholders with different ownership types in the sample of SOEs. The average MOR depth is 8.67%, with a minimum value of 0.03% and a maximum value of 75%, indicating significant differences in the degree of equity balance among nonstate-owned shareholders of SOEs.

## 4.2. Correlation coefficient analysis

To avoid the impact of multicollinearity among variables, this study conducted correlation coefficient analysis on all the variables. Table 3 reports the correlation coefficients between all the variables. Without considering other factors, the mixed-ownership effect is positively correlated with mixed-ownership motivation (Table 3). This finding indicates that the better the company's net profit performance, the better the mixed-ownership effect for enterprises that actively engage in MOR. MOR breadth is positively correlated with its effects, indicating that the greater the MOR breadth, the more types of shareholders there are, and the better the MOR effects. The two indicators of MOR depth are negative, which may be because the average proportion of nonstate-owned shareholders among the top ten shareholders of SOEs in Shandong Province has reached 41.7%. It is not appropriate to increase the shareholding ratio of nonstate-owned shareholders but rather to improve MOR performance by increasing the number of types of shareholders.

From the perspective of the controlling variables, the age of the company listing, growth potential, and financial leverage coefficient are negatively correlated with MOR performance, indicating that the older the company's listing age, i.e., the earlier that the company went public, the worse its MOR performance. This may be due to the accumulation of many criticisms in the long-term operation of the company after listing for a long time, making it difficult to improve performance through MOR at once. The better the growth potential is, the more likely it is that the enterprise is in the maturity stage of its lifecycle and that the MOR effect is not as good as that of enterprises before the maturity stage. The financial leverage coefficient reflects the current financial risk level of the enterprise. The larger the financial leverage coefficient is, the greater the degree of financial risk. When the financial risk of an enterprise is high, MOR performance is worse. The size

**Table 2. Descriptive statistics of the main variables.**

| Variable | Observations | Mean | Median | Var | Min | Max |
|---|---|---|---|---|---|---|
| ROA | 4124 | 0.0311 | 0.0296 | 0.0617 | −0.635 | 0.585 |
| tobinQ | 4124 | 1.632 | 1.256 | 1.248 | 0.625 | 26.82 |
| motive | 4124 | 0.886 | 1 | 0.318 | 0 | 1 |
| extent | 4124 | 2.987 | 3 | 0.710 | 2 | 4 |
| depth | 4124 | 0.0867 | 0.0536 | 0.0943 | 0.0003 | 0.750 |
| HHI | 4124 | 0.819 | 0.856 | 0.127 | 0.206 | 0.991 |
| Age | 4124 | 2.764 | 3.045 | 0.665 | 0 | 3.497 |
| Grow | 4124 | −0.0134 | −0.0666 | 0.493 | −0.990 | 19.88 |
| Size | 4124 | 23.26 | 23.14 | 1.456 | 19.50 | 29.79 |
| Lev | 4124 | 0.505 | 0.512 | 0.206 | 0.0384 | 1.698 |
| Board | 4124 | 2.293 | 2.303 | 0.175 | 1.609 | 2.944 |

**Table 3. Correlation coefficients of the main variables.**

| | ROA | tobinQ | motive | extent | depth | HHI | Age | Grow | Size | Lev | Board |
|---|---|---|---|---|---|---|---|---|---|---|---|
| ROA | 1 | | | | | | | | | | |
| tobinQ | 0.1863* | 1 | | | | | | | | | |
| | 0 | | | | | | | | | | |
| motive | 0.6238* | −0.0277 | 1 | | | | | | | | |
| | 0 | 0.0756 | | | | | | | | | |
| extent | 0.1613* | 0.0484* | 0.0977* | 1 | | | | | | | |
| | 0 | 0.00190 | 0 | | | | | | | | |
| depth | −0.000300 | 0.0368* | −0.0157 | 0.1694* | 1 | | | | | | |
| | 0.985 | 0.0180 | 0.313 | 0 | | | | | | | |
| HHI | −0.1855* | 0.0799* | −0.1671* | −0.0399* | 0.1504* | 1 | | | | | |
| | 0 | 0 | 0 | 0.0105 | 0 | | | | | | |
| Age | −0.0905* | −0.0374* | −0.0460* | 0.0683* | −0.1286* | 0.1822* | 1 | | | | |
| | 0 | 0.0162 | 0.00310 | 0 | 0 | 0 | | | | | |
| Grow | −0.00660 | −0.0315* | −0.0383* | −0.0285 | 0.00110 | 0.0298 | 0.0387* | 1 | | | |
| | 0.673 | 0.0429 | 0.0139 | 0.0669 | 0.945 | 0.0560 | 0.0130 | | | | |
| Size | 0.0839* | −0.3305* | 0.1618* | 0.0954* | 0.0142 | −0.2544* | 0.1043* | −0.0388* | 1 | | |
| | 0 | 0 | 0 | 0 | 0.362 | 0 | 0 | 0.0126 | | | |
| Lev | −0.3614* | −0.2805* | −0.1479* | −0.0676* | −0.0470* | 0.0284 | 0.1152* | 0.0117 | 0.4814* | 1 | |
| | 0 | 0 | 0 | 0 | 0.00250 | 0.0686 | 0 | 0.451 | 0 | | |
| Board | 0.0508* | −0.0408* | 0.0289 | −0.00440 | 0.0301 | 0.0333* | −0.0364* | −0.0281 | 0.2290* | 0.0677* | 1 |
| | 0.00110 | 0.00880 | 0.0636 | 0.777 | 0.0535 | 0.0324 | 0.0194 | 0.0710 | 0 | 0 | |

of the enterprise and the size of the board of directors are positively correlated with the MOR effect, indicating that the larger the enterprise size is, the more standardized the management of the enterprise, and the better the MOR effect. The larger the size of the board of directors, the better the board's governance effect and the better the MOR effect.

In addition, the correlation coefficients between the variables are less than 0.4, indicating that there is no significant multiple correlation phenomenon between the indicators.

### 4.3. Benchmark regression results

Table 4 shows the regression results of the mixed-ownership effect from the perspectives of extent and depth (HHI), mixed-ownership motivation, and the cross term between mixed-ownership motivation and breadth and depth. When the degree of mixed-ownership among heterogeneous shareholders in the MOR of SOEs is measured via extra and the degree of combination among heterogeneous shareholders in the MOR of SOEs is measured via the HHI, the regression results in Column (1) reveal that in the absence of control variables, the positive effect of shareholder diversity (extra) on the MOR effect of SOEs is not significant; however, the positive effect of depth (HHI) on the mixed effect is significant at the 5% level. The second column shows that with the addition of control variables, the impact of mixed-modification breadth on the mixed-modification effect is significantly positively correlated at the 1% level, indicating that the larger the mixed-modification breadth is, the better the mixed labeling effect. Hypothesis H1 is thus validated. The impact of mixed-modification depth on the mixed-modification effect is significantly positively correlated at the 1% level, indicating that the greater the mixed-modification depth, the better the mixed-modification effect. Hypothesis H2 is therefore supported. In addition, enterprise size and the MOR effect are significantly positively correlated with respect to the control variables, whereas the financial leverage coefficient is significantly negatively correlated with the MOR effect. The third

**Table 4. Benchmark regression results.**

| VARIABLES | (1) ROA | (2) ROA | (3) ROA | (4) ROA |
|---|---|---|---|---|
| extent | 0.0105 | 0.0072*** | 0.0047*** | 0.0110*** |
| | (0.817) | (5.310) | (4.231) | (3.172) |
| HHI | 0.0864** | 0.0430*** | 0.0227*** | 0.0895*** |
| | (2.115) | (4.285) | (2.743) | (4.135) |
| motive | | | 0.0938*** | 0.0511** |
| | | | (31.400) | (2.401) |
| motive* extent | | | | 0.0071* |
| | | | | (1.936) |
| motive *HHI | | | | 0.0716*** |
| | | | | (3.212) |
| Age | | −0.0064*** | −0.0055*** | −0.0056*** |
| | | (−3.555) | (−3.943) | (−4.030) |
| Grow | | −0.0014 | −0.0010 | −0.0008 |
| | | (−0.467) | (−0.465) | (−0.356) |
| Size | | 0.0110*** | 0.0056*** | 0.0056*** |
| | | (8.791) | (5.336) | (5.361) |
| Lev | | −0.1295*** | −0.0829*** | −0.0822*** |
| | | (−16.181) | (−13.326) | (−13.161) |
| Board | | 0.0133** | 0.0120** | 0.0123** |
| | | (1.993) | (2.198) | (2.256) |
| Constant | 0.0613*** | −0.1655*** | −0.1447*** | −0.1045*** |
| | (3.355) | (−5.447) | (−5.846) | (−3.340) |
| YEAR | control | control | control | control |
| IND | control | control | control | control |
| Observations | 4, 124 | 4, 124 | 4, 124 | 4, 124 |
| R-squared | 0.173 | 0.327 | 0.592 | 0.595 |

column adds the independent variable of mixed-ownership motivation (motivation) on the basis of the second column. The regression results show that mixed-ownership motivation is significantly positively correlated with the mixed-ownership effect at the 1% level; that is, the more the mixed-ownership motivation tends toward active mixed-ownership, that is, survival needs, the better the mixed-ownership effect. The fourth column shows the regression results of all the variables, which include the interaction terms of mixed-ownership motivation and extra mixed-ownership motivation and HHI on the basis of (3). The regression results indicate that mixed-ownership breadth, mixed-ownership depth, mixed-ownership motivation, and crossover terms are significantly positively correlated with the mixed-ownership effect. The cross term between MOR breadth and the motivation for MOR is significantly positively correlated, indicating that, as the motivation for active MOR (survival needs) increases, MOR breadth increases, the number of shareholders increases, and the MOR effect improves. Conversely, as the motivation for passive MOR (policy push) increases, MOR breadth and MOE depth increase, and the MOR effect deteriorates, supporting Hypotheses H3a and H3b.

## 4.4. Robustness tests

**4.4.1. Replacing the dependent variable.** This article draws on the research of Hu Renkun (2024) [19] and selects Tobin's Q to measure the mixed effect again. The regression results are shown in Table 5. The regression results

**Table 5. Regression results after replacing the dependent variable.**

| VARIABLES | (1) tobinQ | (2) tobinQ | (3) tobinQ | (4) tobinQ |
|---|---|---|---|---|
| extent | 0.0388 | 0.0820*** | 0.0819*** | 0.0141** |
| | (1.272) | (3.979) | (3.974) | (2.247) |
| HHI | 0.1789 | 0.2492* | 0.2487* | 0.2060*** |
| | (0.960) | (−1.745) | (1.738) | (3.519) |
| motive | | | 0.0024** | 0.1703*** |
| | | | (2.150) | (6.456) |
| motive *HHI | | | | 0.0464* |
| | | | | (1.711) |
| motive* extent | | | | 0.0758* |
| | | | | (1.751) |
| Age | | 0.0224 | 0.0224 | 0.0225 |
| | | (0.655) | (0.657) | (0.657) |
| Grow | | −0.2911*** | −0.2911*** | −0.2924*** |
| | | (−5.931) | (−5.932) | (−5.944) |
| Size | | −0.2281*** | −0.2282*** | −0.2285*** |
| | | (−6.919) | (−6.914) | (−6.913) |
| Lev | | −0.5386*** | −0.5374*** | −0.5377*** |
| | | (−3.072) | (−3.030) | (−3.023) |
| Board | | 0.1166 | 0.1165 | 0.1162 |
| | | (0.753) | (0.753) | (0.751) |
| Constant | 2.4818 | 7.6515 | 7.6521 | 7.8084 |
| | (1.003) | (0.230) | (1.184) | (1.041) |
| YEAR | control | control | control | control |
| IND | control | control | control | control |
| Observations | 4124 | 4124 | 4124 | 4124 |
| R-squared | 0.279 | 0.363 | 0.354 | 0.376 |

Note: * * * represents p < 0.01, * * represents p < 0.05, and * represents p < 0.1. The t test values are in parentheses.

reveal that without control variables, the relationships between mixed-change breadth and mixed-change depth; the mixed-change effect is not significant. After the control variables are added, the two variables are significantly positively correlated with the mixed-change effect. The full variable regression results are shown in Column (4), where the interaction terms between mixed-change motivation and mixed change breadth and the interaction terms between mixed-change motivation and mixed-change depth are significantly positively correlated. The conclusion is not substantially changed.

**4.4.2. Replacing the independent variable.** This study draws on the approach of Cao Yue et al. (2020), where the independent variable mixed-ownership depth is regressed, using the sum of the shareholding ratios of nonstate-owned shareholders among the top ten shareholders of mixed-ownership SOEs as the proxy variable for mixed-ownership depth and tests the impact of mixed-ownership depth on the mixed-ownership effect. The regression results are shown in Table 6, and the main effect has not changed.

In addition, to test the impact of mixed-ownership motivation on the mixed-ownership effect, this study grouped the samples according to mixed-ownership motivation. The sample with mixed-ownership motivation for survival needs is

**Table 6. Replace independent variables and perform robustness testing regression after grouping.**

| Variables | (1) ROA | (2) ROA | (3) ROA | (4) ROA | (5) ROA Active | (6) ROA Passive |
|---|---|---|---|---|---|---|
| extent | 0.0111*** | 0.0078*** | 0.0049*** | 0.0129*** | 0.0038*** | 0.0079** |
| | (6.817) | (5.575) | (4.278) | (3.534) | (3.244) | (2.157) |
| depth | 0.0037 | 0.0256** | 0.0079 | 0.0839** | 0.0022*** | 0.0570* |
| | (0.250) | (1.988) | (0.822) | (2.433) | (6.232) | (1.760) |
| motive | | | 0.0943*** | 0.1118*** | | |
| | | | (31.384) | (10.090) | | |
| motive *depth | | | | 0.0866** | | |
| | | | | (2.437) | | |
| motive* extent | | | | 0.0089** | | |
| | | | | (2.365) | | |
| Age | | −0.0087*** | −0.0065*** | −0.0066*** | −0.0065*** | −0.0022 |
| | | (−4.901) | (−4.867) | (−4.898) | (−4.823) | (−0.540) |
| Grow | | −0.0016 | −0.0012 | −0.0009 | −0.0018 | 0.0050 |
| | | (−0.529) | (−0.518) | (−0.402) | (−0.782) | (0.785) |
| Size | | 0.0126*** | 0.0063*** | 0.0062*** | 0.0053*** | 0.0169*** |
| | | (10.309) | (6.396) | (6.352) | (4.963) | (7.304) |
| Lev | | −0.1346*** | −0.0852*** | −0.0849*** | −0.0842*** | −0.0843*** |
| | | (−16.891) | (−13.931) | (−13.825) | (−12.685) | (−6.220) |
| Board | | 0.0099 | 0.0101* | 0.0099* | 0.0109* | −0.0188 |
| | | (1.511) | (1.900) | (1.858) | (1.946) | (−1.366) |
| Constant | −0.0104 | −0.2171*** | −0.1714*** | −0.1853*** | −0.0580** | −0.3382*** |
| | (−0.597) | (−7.527) | (−7.617) | (−7.718) | (−2.201) | (−6.528) |
| YEAR | control | control | control | control | control | control |
| IND | control | control | control | control | control | control |
| Observations | 4124 | 4124 | 4124 | 4124 | 3653 | 471 |
| R-squared | 0.138 | 0.321 | 0.590 | 0.593 | 0.317 | 0.412 |

Note: *** represents p<0.01, ** represents p<0.05, and * represents p<0.1. The t test values are in parentheses.

termed active mixed-ownership, and the sample with mixed-ownership motivation for policy promotion is termed passive mixed ownership. The regression results after grouping are shown in Columns (5) and (6) of Table 6. The signs of the main variables in the active mixing group do not change but are more significant. The signs of the passive mixing group also do not change, but their significance decreases significantly. Further evidence shows that the motivation for MORs has a significant effect on the MOR effect.

## 5. Further research

### 5.1. Adjustment effect test based on shareholder correlations

According to the foregoing, the degree of deepening of the MOR of SOEs and its promoting effect on the effectiveness of the reform depend on the effectiveness of the mutual supervision and restraint mechanism between nonstate-owned shareholders. When nonstate-owned shareholders participate in investing in SOEs and have a connection with state-owned shares, this weakens their supervisory and constraining role in the possible infringement of their own interests by controlling shareholders. This study follows the approach of Cao Yue et al. (2020) and sets a dummy variable If_Selated.

When nonstate-owned investors in non-SOEs are associated with state-owned shares, the If_Selated value is set to 1; otherwise, it is 0. The relevant information is sourced from the CSMAR database, which publicly discloses the identification information of shareholders and their affiliated shareholders of listed companies. The contact patterns between shareholders and their affiliated parties include but are not limited to ① blood relationships, ② joint actors, ③ shareholding associations, ④ direct correlation, ⑤ entrusted operations, ⑥ authorized operations, and ⑦ other forms. This article first identifies whether the company investors disclosed in the database are among the top ten investors of listed companies, then verifies whether the affiliated investors of the investor are also among the top ten investors, and, finally, evaluates the shareholding attributes of the affiliated investors. If the affiliated owners corresponding to the state owner are not within the state-owned category, there is a relationship between state-owned capital and nonstate-owned owners among the top ten owners of the enterprises.

Table 7 shows the regression results considering shareholder associations. The data in the table indicate that the interaction term If_Selated * HHI is significantly negative at least at the 5% level, indicating that when nonstate-owned shareholders who participate in equity are associated with state-owned shares, the positive effect of MOR depth on the MOR

**Table 7. Regression results considering shareholder correlations.**

| Variables | (1) ROA | (2) ROA | (3) ROA | (4) ROA |
|---|---|---|---|---|
| extent | 0.0108*** | 0.0074*** | 0.0075*** | 0.0075*** |
|  | (7.004) | (5.420) | (5.459) | (5.428) |
| HHI | 0.0601*** | 0.0249*** | 0.0248*** | 0.0257*** |
|  | (7.302) | (3.198) | (3.189) | (3.234) |
| If_Related |  |  | −0.0045 | −0.0042** |
|  |  |  | (−1.262) | (−2.247) |
| If_Related *HHI |  |  |  | −0.0164*** |
|  |  |  |  | (3.968) |
| If_Related * extent |  |  |  | −0.0014 |
|  |  |  |  | (−0.235) |
| Age |  | −0.0067*** | −0.0068*** | −0.0068*** |
|  |  | (−3.786) | (−3.797) | (−3.792) |
| Grow |  | −0.0009 | −0.0009 | −0.0009 |
|  |  | (−0.310) | (−0.310) | (−0.312) |
| Size |  | 0.0113*** | 0.0113*** | 0.0113*** |
|  |  | (8.964) | (8.955) | (8.914) |
| Lev |  | −0.1309*** | −0.1310*** | −0.1310*** |
|  |  | (−16.267) | (−16.283) | (−16.278) |
| Board |  | 0.0120* | 0.0121* | 0.0122* |
|  |  | (1.798) | (1.825) | (1.839) |
| Constant | 0.0393** | −0.1823*** | −0.1823*** | −0.1815*** |
|  | (2.236) | (−5.983) | (−5.984) | (−5.948) |
| YEAR | control | control | control | control |
| IND | control | control | control | control |
| Observations | 4124 | 4124 | 4124 | 4124 |
| R-squared | 0.166 | 0.324 | 0.324 | 0.324 |

Note: * * * represents p < 0.01, * * represents p < 0.05, and * represents p < 0.1. The t test values are in parentheses.

effect is significantly weakened. However, the interaction term If_Selated * extreme is not significant, indicating that the positive impact of shareholder association on the breadth and effectiveness of MORs is not significant. The above results indicate that the independence and supervisory balance between nonstate-owned shareholders and state-owned shareholders are important factors affecting the effectiveness of the MOR of SOEs.

### 5.2. Testing the regulatory effect based on the degree of marketization

Given the differences in institutional backgrounds and market openness across different regions of China, their degree of government regulation also shows an uneven state [20]. On the other hand, when the institutional environment in the region where SOEs are located is poor and government intervention increases, the role of accounting indicators in their performance evaluation tends to weaken [11]. At this point, nonstate-owned shareholders who participate in investment are more likely to play a role in improving the rigid management mechanism of SOEs [12]. The diversified equity structure generated by the MOR of SOEs can help alleviate internal control difficulties dominated by state-owned shares and can play a reinforcing role in situations where the external institutional framework is still insufficient. On the other hand, when the institutional environment in the region where SOEs are located is more favorable, the level of government intervention is lower, and the competition between markets becomes more intense, enterprises are more likely to rely on conventional operations and strengthen internal governance mechanisms to achieve profit goals. In this context, the diversified equity structure constructed by the MOR of SOEs has a relatively small effect on enhancing the effectiveness of the reform. On the basis of the above analysis, it can be inferred that MORs help offset the impact of incomplete external institutional frameworks on SOEs. In regions with lower levels of market development, the MOR of SOEs practice can significantly optimize the reform results. This article refers to Liu Xiaoli et al.'s (2024) [50] marketization index to measure the state of marketization (Market). The regression analysis in Table 8 reveals that the product of Market and HHI is negative at a significance level of 5%. This finding indicates that in regions with lower levels of marketization, the positive effect of MOR depth on reform effectiveness is more significant. In addition, although the Market*extent is negative and not significant, this reveals that the deepening of the marketization process does not significantly change the correlation between shareholder diversity and the effectiveness of the MOR. The conclusions above indicate that the mixed-ownership pattern generated by the MOR within SOEs can help alleviate the negative impact of inadequate external market mechanisms on the effectiveness of the reform.

## 6. Conclusions and limitations

This study empirically investigates the impact of mixed-ownership reform (MOR) on the performance of Chinese state-owned enterprises (SOEs), with a specific focus on the roles of reform characteristics (breadth and depth) and the critical moderating effect of reform motivations. On the basis of a panel dataset of listed SOEs from 2013 to 2023, our findings yield three key conclusions:

The motivation behind MORs is a fundamental determinant of their success. Reforms that are survival driven (initiated by SOEs facing performance distress) demonstrate a strong positive relationship between the breadth/depth of reform and subsequent performance improvements. In contrast, policy-driven reforms (often adopted by already profitable SOEs, primarily to comply with top-down mandates) show a negligible or even negative effect from increasing reform dimensions. This underscores that the intrinsic need for change is more critical than the external imposition of reform structures.

The efficacy of introducing nonstate shareholders is contingent upon their independence. Our evidence indicates that affiliated transactions or close relationships between nonstate and state shareholders significantly undermine the positive governance effects of MORs. This finding highlights that mere diversification of equity is insufficient; the genuine independence of nonstate shareholders is a prerequisite for effective checks and balances and the mitigation of insider control.

Moreover, the institutional context matters. The positive impact of MOR, particularly its depth, is more pronounced in regions with lower levels of marketization. This suggests that a diversified equity structure can serve as a substitute

**Table 8. Regression results considering the degree of marketization.**

| Variables | (1) ROA | (2) ROA | (3) ROA | (4) ROA |
|---|---|---|---|---|
| extent | 0.0108*** | 0.0074*** | 0.0073*** | 0.0166** |
|  | (7.004) | (5.420) | (5.405) | (2.138) |
| HHI | 0.0601*** | 0.0249*** | 0.0248*** | 0.0798* |
|  | (7.302) | (3.198) | (3.198) | (1.771) |
| market |  |  | 0.0007 | 0.0008 |
|  |  |  | (0.823) | (0.211) |
| Market*HHI |  |  |  | −0.0055** |
|  |  |  |  | (−2.264) |
| Market*extent |  |  |  | −0.0009 |
|  |  |  |  | (−1.215) |
| Age |  | −0.0067*** | −0.0067*** | −0.0068*** |
|  |  | (−3.786) | (−3.749) | (−3.839) |
| Grow |  | −0.0009 | −0.0009 | −0.0007 |
|  |  | (−0.310) | (−0.297) | (−0.240) |
| Size |  | 0.0113*** | 0.0113*** | 0.0114*** |
|  |  | (8.964) | (8.963) | (8.993) |
| Lev |  | −0.1309*** | −0.1302*** | −0.1299*** |
|  |  | (−16.267) | (−15.994) | (−15.995) |
| Board |  | 0.0120* | 0.0120* | 0.0118* |
|  |  | (1.798) | (1.799) | (1.779) |
| Constant | 0.0393** | −0.1823*** | −0.1876*** | −0.1703*** |
|  | (2.236) | (−5.983) | (−6.192) | (−3.471) |
| YEAR | control | control | control | control |
| IND | control | control | control | control |
| Observations | 4124 | 4124 | 4124 | 4124 |
| R-squared | 0.166 | 0.324 | 0.324 | 0.325 |

Note: * * * represents p < 0.01, * * represents p < 0.05, and * represents p < 0.1. The t test values are in parentheses.

mechanism for underdeveloped external market institutions, helping to alleviate the inefficiencies associated with excessive government intervention.

While this study provides valuable insights, several limitations should be acknowledged, suggesting fruitful directions for future research. Our classification of reform motivations based on prereform profitability, while theoretically grounded and empirically practical, is a proxy. It may not fully capture the nuanced internal deliberations of SOE managers. Future research could employ qualitative methods, such as in-depth case studies or surveys, to develop a more granular understanding of the decision-making processes behind MORs. The findings are based on listed SOEs, which are generally larger and operate under stricter disclosure rules than nonlisted SOEs do. The applicability of our conclusions to the broader population of nonlisted SOEs, which may face different constraints and governance issues, remains an open question. Future studies could explore MOR effects in this important segment. This study focuses on regional marketization as a key contextual factor. Future research could investigate the influence of other moderating variables, such as industry-specific characteristics, the nature of the ultimate controller, or the composition of the board of directors, to construct a more comprehensive contingency model of MOR effectiveness.

## Supporting information

**S1 File. Contains 10 original Excel datasets that support the analyses presented in the article.** The files correspond to the following variables and themes (all data are structured in standard spreadsheet format for reproducibility): **balance:** Dataset related to firms' balance sheet indicators (e.g., assets, liabilities, and owner's equity). board: Data on corporate board characteristics (e.g., board size). **company:** Basic information of sample companies (e.g., Date of establishment, Date of listing). **EquityNature+English:** Dataset on the nature of corporate equity (e.g., ownership type, state-owned vs. non-state-owned shares). **income+English:** Data on firms' income statement metrics (e.g., Operating Cost, net profit, operating income). **market+English:** Market-related indicators (e.g., Marketization Index of the Region Where Listed Companies Are Located). **profit_ratio:** Dataset of profitability ratios (e.g., return on assets, return on equity, EBIT). **relation+English:** Shareholder Related Relationship Data (e.g., Relationship Level Identifier, Related Party (Shareholder) Name, Whether it is a Direct Shareholder, Whether it is an Indirect Shareholder, Shareholding Ratio). **shareholder+English:** Shareholder Shareholding Information Data (e.g., Shareholding Rank, Shareholding Quantity, Shareholding Ratio). **tobinq:** Dataset of Tobin's Q values (a measure of firm value, calculated as the ratio of market value to book value of assets).
(ZIP)

## Author contributions

**Conceptualization:** Xiuting Wang.

**Data curation:** Xiuting Wang.

**Formal analysis:** Xiuting Wang.

**Funding acquisition:** Xiuting Wang.

**Investigation:** Xiuting Wang.

**Methodology:** Xiuting Wang.

**Project administration:** Xiuting Wang.

**Resources:** Xiuting Wang.

**Software:** Xiuting Wang.

**Supervision:** Xiuting Wang.

**Validation:** Xiuting Wang.

**Visualization:** Xiuting Wang.

**Writing – original draft:** Xiuting Wang.

**Writing – review & editing:** Xiuting Wang.

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
