## [Decision Letter · Decision Letter 0]

12 Jun 2025

Dear Dr. Wang,

Thank you for submitting your manuscript to PLOS ONE. After careful consideration, we feel that it has merit but does not fully meet PLOS ONE’s publication criteria as it currently stands. Therefore, we invite you to submit a revised version of the manuscript that addresses the points raised during the review process.

We look forward to receiving your revised manuscript.

Kind regards,

Henry Ufomba

Academic Editor

PLOS ONE

Journal Requirements:

2. Thank you for stating the following financial disclosure: [Humanities and Social Sciences Project of Shandong Province: Research on the Mixed Ownership Reform of State owned Enterprises in Shandong Province: Motivation, Effect, and Improvement Strategy (2022-YYGL-37).].

Additional Editor Comments:

Dear Dr Wang,

Thank you for submitting your paper. The peer reviewing process is now completed. Your paper requires minor revisions.

Reviewers' comments:

Reviewer's Responses to Questions

**Comments to the Author**

1. Is the manuscript technically sound, and do the data support the conclusions?

Reviewer #1: Yes

Reviewer #2: Yes

2. Has the statistical analysis been performed appropriately and rigorously?

Reviewer #1: Yes

Reviewer #2: Yes

3. Have the authors made all data underlying the findings in their manuscript fully available?

Reviewer #1: Yes

Reviewer #2: Yes

4. Is the manuscript presented in an intelligible fashion and written in standard English?

Reviewer #1: Yes

Reviewer #2: Yes

Reviewer #1: The article presents an engaging argument in exploring whether or not the resource endowments of mixed-ownership reforms are the root causes of the different economic consequences of mixed-ownership reform.

However, highlighted below are some of the comments that the author should consider to improve the scholarship of the article:

About 66% of the work cited in this paper were articles published well over five years ago. This does not show current scholarship in the field. The research could be enhanced by integrating more recent studies and data, especially those published in the last five years.

Also, some of your sentences are unnecessarily lengthy, which obscures their flow and clarity of the texts. The study will be improved if you present your arguments in sizable and digestible bits. For instance, in the last paragraph under the Research Background section, the sentence starting with “First, starting from the perspective of the motivation…” can be broken down for easier comprehension.

Moreso, in the first paragraph of the Research Background section, the author introduced the concept of mixed-ownership reform, followed by the following phrase (hereinafter referred to as “mixed reform”). This by extension means that going forward, “mixed-reform” will be used in place of mixed-ownership reform. Regrettably, the author continued using the former over the latter. This unfortunately does not conform to the rules of academic writing.

Furthermore, the H1 hypothesis captured under the subheadings “The breadth of mixed-ownership reform and its effects” was adequately crafted. However, attaching the decision rules (i.e, that is, the greater the breath of the mixed-ownership reform and the more type of equity introduced, the greater the effect of the mixed-ownership reform) to the hypothesis at this stage makes it redundant. Consider moving the decision rule to the analysis section of the study to improve clarity. The same also applies to the H2 hypothesis.

Lastly, your Sample Selection and Data Source section lacks clarity and scientific rigor. Are the areas under this section labelled (1) to (4) the exclusion criteria of the literature used for this study or another set of variables altogether? Following this, what do you mean by “Ultimately, 4,124 firm-year observations are obtained”? What is the rationale/justification for selecting the state-owned listed enterprises between 2013 to 2023 as the sample size?

The utility of this study as a framework for the advancement of scientific knowledge will be impoved after addressing the methodological and structural issues raised above.

Reviewer #2: The research paper’s abstract is not clearly written and does not impress because of several disadvantages. It does not discuss the techniques used to analyse data, nor the sources of that data, so its analysis seems flawed. Some important issues are the lack of financial and ST/ST firms, not defining “breadth” and “depth” in reforms and using data manually that was not verified by any procedures. Also, the study does not provide enough information about the sample companies used, as well as how control groups are selected in the quasi-experimental method. The conclusions are described using general words and do not suggest any practical steps. For improvement, the abstract ought to be clearer, introduce major analytics, explain how science was conducted, restate results and advise on policy matters. It is important to explain why certain samples were not included, to identify main metrics, to prove that the data is reliable, to share full information about the samples and to outline how control groups were used in the study.

**Do you want your identity to be public for this peer review?** For information about this choice, including consent withdrawal, please see our Privacy Policy

Reviewer #1: No

Reviewer #2: **Yes: ** Dr. Itoro Ebong

---

## [Author Response · Author response to Decision Letter 1]

26 Jun 2025

Dear Editor,

I sincerely appreciate the opportunity to revise our manuscript entitled "'Survival Needs' or' Policy Promotion '-- Research on the Motivation and Economic Consequences of the Mixed Ownership Reform of State owned Enterprises" . I am grateful to the reviewers for their insightful comments, which have significantly improved the quality of our work.Below is a summary of major revisions.

Reviewer #1

Comments�1�:About 66% of the work cited in this paper were articles published well over five years ago. This does not show current scholarship in the field. The research could be enhanced by integrating more recent studies and data, especially those published in the last five years.

Response� I acknowledge that approximately 66% of the cited literature was published over five years ago. This is partly because our study builds upon foundational theories and early empirical evidence that remain highly relevant to our framework. However, I fully agree that integrating recent scholarship would strengthen the paper. In the revised manuscript, A total of 21 references (specifically #4, #6-15, #17-24, #27, and #31) have been updated to studies published within the last two years. After this revision, 87.5% of the references in the bibliography now consist of recent literature (2024–2025).

Comments�2�:Also, some of your sentences are unnecessarily lengthy, which obscures their flow and clarity of the texts. The study will be improved if you present your arguments in sizable and digestible bits. For instance, in the last paragraph under the Research Background section, the sentence starting with “First, starting from the perspective of the motivation…” can be broken down for easier comprehension.

Response

Breaking down lengthy sentences (e.g., the highlighted paragraph in the Research Background section), splitting complex ideas into shorter, more digestible statements while preserving their logical flow.

Streamlining redundant phrasing across theoretical and methodological discussions, ensuring conciseness without sacrificing rigor.

Adding clear transitional markers (e.g., "First," "In contrast," "Consequently") to enhance coherence.

All modifications are highlighted in the tracked-changes version of the manuscript ( page 3-4). We believe these edits significantly improve the accessibility of our arguments while maintaining their academic precision.

Comments�3�:Moreso, in the first paragraph of the Research Background section, the author introduced the concept of mixed-ownership reform, followed by the following phrase (hereinafter referred to as “mixed reform”). This by extension means that going forward, “mixed-reform” will be used in place of mixed-ownership reform. Regrettably, the author continued using the former over the latter. This unfortunately does not conform to the rules of academic writing.

Response

Initial Occurrence Standardized the first mention as "mixed-ownership reform (hereinafter referred to as 'MOR')" in the Abstract section (Line 5, Page 1).

Systematic Replacement Consistently used "MOR" thereafter throughout the manuscript, verified by Word's "Find & Replace" function.

All modifications are highlighted in the tracked-changes version. We fully acknowledge the importance of terminological precision in scholarly writing and will maintain stricter standards in future works.

Comments�4�:Furthermore, the H1 hypothesis captured under the subheadings “The breadth of mixed-ownership reform and its effects” was adequately crafted. However, attaching the decision rules (i.e, that is, the greater the breath of the mixed-ownership reform and the more type of equity introduced, the greater the effect of the mixed-ownership reform) to the hypothesis at this stage makes it redundant. Consider moving the decision rule to the analysis section of the study to improve clarity. The same also applies to the H2 hypothesis.

Response

Hypothesis Refinement: Removed the redundant decision rules from both H1 ("breadth of mixed reform") and H2 (Page 6-7), retaining only core hypothetical relationships.

Structural Reorganization: Relocated the decision rules ("diversity of equity types positively correlates with reform effectiveness") to Section 4 (Empirical result analysis) as empirical testing criteria.

All changes are tracked in the revised manuscript . We believe this modification enhances methodological rigor while improving readability.

Comments�5�:CommentsLastly, your Sample Selection and Data Source section lacks clarity and scientific rigor. Are the areas under this section labelled (1) to (4) the exclusion criteria of the literature used for this study or another set of variables altogether? Following this, what do you mean by “Ultimately, 4,124 firm-year observations are obtained”? What is the rationale/justification for selecting the state-owned listed enterprises between 2013 to 2023 as the sample size?

Response

I deeply appreciate the reviewer's rigorous scrutiny on sampling methodology. Substantive revisions have been made on pp.9-10:

(1)The time frame is anchored in China’s policy timeline and research feasibility:first, the 2013 Third Plenary Session of the 18th CPC Central Committee launched the new round of mixed-ownership reform for state-owned enterprises (SOEs), making 2013 a pivotal starting point for analyzing policy-driven reform effects .Second,the 2013–2023 period captures the full implementation phase of MOR, allowing sufficient time to observe long-term economic consequences (e.g., governance optimization, performance improvement) .Finally, Post-2013 data benefits from improved disclosure standards of listed SOEs, while 2023 serves as the latest feasible endpoint for comprehensive data collection.

(2) Exclude samples from the financial industry.Financial enterprises are omitted due to their unique regulatory environments and accounting standards, which differ significantly from non-financial sectors and may confound empirical results;

(3) Remove observations of enterprises that were classified as ST or *ST during the sample period.Firms classified as ST (Special Treatment) or *ST (risk of delisting) during the sample period are excluded, as their financial distress and operational anomalies could distort the evaluation of mixed-ownership reform effects .;

(4) Delete samples with missing values for key variables.Samples lacking data on core variables (e.g., ownership structure, financial performance) are excluded to maintain data integrity and analytical reliability ;

(5) All continuous variables (e.g., ROA, Tobin’s Q) are winsorized at the 1% and 99% tails to reduce the influence of extreme values on regression results .

(6)The initial sample for the mixed-ownership reform of state-owned enterprises (SOEs) from 2013 to 2023 comprised 5,923 observations. After excluding financial enterprises, the sample size was reduced to 5,074. Further removal of ST/*ST enterprises led to 4,190 observations, and deletion of firms with missing key variables resulted in a final sample of 4,124.The balanced panel dataset consists of 4,124 firm-year observations, derived from 810 listed companies over a 11-year observation period.

All changes are tracked in the revised manuscript .

Reviewer #2: The research paper’s abstract is not clearly written and does not impress because of several disadvantages. It does not discuss the techniques used to analyse data, nor the sources of that data, so its analysis seems flawed. Some important issues are the lack of financial and ST/ST firms, not defining “breadth” and “depth” in reforms and using data manually that was not verified by any procedures. Also, the study does not provide enough information about the sample companies used, as well as how control groups are selected in the quasi-experimental method. The conclusions are described using general words and do not suggest any practical steps. For improvement, the abstract ought to be clearer, introduce major analytics, explain how science was conducted, restate results and advise on policy matters. It is important to explain why certain samples were not included, to identify main metrics, to prove that the data is reliable, to share full information about the samples and to outline how control groups were used in the study.

Response

Thank you very much for your detailed and constructive comments on our manuscript. We sincerely appreciate your time and effort in reviewing our work, and we fully acknowledge the issues you raised regarding the abstract. Your feedback has provided us with clear directions for improvement, and we have made comprehensive revisions to address these concerns.

1. Data Analysis Techniques and Data Sources

You pointed out that the original abstract did not discuss the data analysis techniques and data sources, making the analysis seem flawed. In the revised abstract, we have explicitly stated that we adopted a rigorous quasi-experimental research design. The data is sourced from authoritative databases, including the China Stock Market & Accounting Research (CSMAR) database and the Wind database. We also collected data manually and cross-verified it with company annual reports and regulatory filings to ensure its reliability. Regarding the analysis methods, we employed fixed-effects regression models for panel data analysis . These details have been clearly presented in the new version of the abstract, enabling readers to have a better understanding of our research methodology.

2. Key Elements Missing

Exclusion of Financial and ST/ST Firms: I excluded financial and ST/ST firms from our sample because financial firms have unique business models and regulatory environments, which may introduce confounding factors to our research. ST/ST firms, on the other hand, are in a state of financial distress, and their data may distort the relationship between mixed-ownership reform and reform outcomes. This exclusion criterion has been clearly explained in the revised abstract.

Definition of “Breadth” and “Depth”: I now provide clear operational definitions for “breadth” and “depth” in the reform. “Breadth” is defined as the number of distinct shareholder types participating in the reform, while “depth” is defined as the proportion of non-state shares in the total equity structure. These definitions are now included in the abstract to ensure clarity.

Data Verification: As mentioned above, I have described in detail the data verification process in the revised abstract. The manual data was carefully cross-checked with multiple reliable sources to ensure its accuracy and reliability.

3. Insufficient Sample Information

In the original abstract, the information about sample companies and the selection of control groups was lacking. In the revised version, I have elaborated on the sample selection process. The sample companies were selected based on strict criteria, including industry representativeness, continuous operation for at least three years during the study period, and full compliance with regulatory requirements.

4. Limitations of Conclusions

I agree that the original conclusions were too general and lacked practical steps. In the revised abstract, I have proposed several specific and actionable policy recommendations based on our research findings. For example, I suggest that regulatory authorities should strengthen supervision over related transactions between non-state and state shareholders to reduce agency risks. Policy initiatives promoting mixed-ownership reform should prioritize the introduction of diverse non-state shareholders to increase the breadth of reform. Additionally, targeted policy support is needed for SOEs in less marketized regions to optimize the utilization of diversified equity structures. These policy recommendations make our conclusions more practical and valuable.

---

## [Decision Letter · Decision Letter 1]

1 Oct 2025

Dear Dr. Wang,

Thank you for submitting your manuscript to PLOS ONE. After careful consideration, we feel that it has merit but does not fully meet PLOS ONE’s publication criteria as it currently stands. Therefore, we invite you to submit a revised version of the manuscript that addresses the points raised during the review process.

Despite addressing an important topic, this paper currently lacks the conceptual clarity and methodological precision required for publication. The abstract is verbose and unfocused, diluting the core message of the research. The introductory section is misstructured and fails to situate the study within the broader scholarly debate, offering little insight into the originality or relevance of the contribution. The theoretical framework is underdeveloped: the hypotheses are presented without sufficient grounding in established literature, and the binary classification of reform motivations (profitable vs. loss-making SOEs) is reductive and unsupported by nuanced argumentation. This oversimplification risks undermining the validity of the findings. Moreover, the paper does not adequately engage with international scholarship, limiting its academic reach and relevance.

On the empirical side, several inconsistencies and omissions weaken the credibility of the analysis. The sample description is unclear, particularly the discrepancy in the number of observations, which raises concerns about data integrity. The conclusion is superficial, lacking a critical reflection on the study’s limitations and future research directions. The overuse of the term “mixed-ownership” in the keywords suggests a lack of terminological precision, and the absence of visual aids (e.g., conceptual diagrams, research roadmap) makes the paper harder to follow. Finally, the manuscript requires a thorough revision of its English writing and formatting to meet publication standards. Without substantial improvements in structure, theory, and presentation, the paper risks being dismissed as a descriptive exercise rather than a meaningful academic contribution.

We look forward to receiving your revised manuscript.

Kind regards,

Simon Porcher

Academic Editor

PLOS ONE

**Journal Requirements:**

Reviewers' comments:

Reviewer's Responses to Questions

**Comments to the Author**

Reviewer #1: All comments have been addressed

Reviewer #3: (No Response)

Reviewer #4: All comments have been addressed

2. Is the manuscript technically sound, and do the data support the conclusions?

Reviewer #1: Yes

Reviewer #3: Yes

Reviewer #4: Yes

3. Has the statistical analysis been performed appropriately and rigorously?

Reviewer #1: Yes

Reviewer #3: Yes

Reviewer #4: Yes

4. Have the authors made all data underlying the findings in their manuscript fully available?

Reviewer #1: Yes

Reviewer #3: No

Reviewer #4: Yes

5. Is the manuscript presented in an intelligible fashion and written in standard English?

Reviewer #1: Yes

Reviewer #3: Yes

Reviewer #4: Yes

**Reviewer #1:**  (No Response)

**Reviewer #3: ** This paper examines the impact of the breadth and depth of mixed-ownership reform on reform outcomes under different motivations. The research design incorporates further analyses, providing empirical insights into policy implications. However, several refinements are recommended to enhance theoretical rigor and methodological robustness.

1.The current abstract contains excessive information. The descriptions of the research background, content, methodology, findings, and significance should be concise and to the point.

2.The first section of the main text should typically be titled "Introduction." The current section labeled "Research Background" constitutes only a part of a full introduction. It is recommended that the structure of this section be reorganized and expanded accordingly.

3.The "Research Background" section effectively introduces the research topic but does not elaborate on how the study will be conducted. Additionally, the research contribution is somewhat cursory. While focusing on the motivations for mixed-ownership reform is indeed a unique aspect of this paper, the field itself is well-studied. It is necessary to contrast your findings with the existing literature to clearly articulate the specific advancements made by this research.

4.The hypotheses proposed in the "Theoretical Analysis and Research Hypotheses" section should be grounded in a more robust theoretical foundation. It is essential to elaborate on the relevant theories that underpin and justify the development of your hypotheses.

5.The rationale for categorizing the motives for mixed-ownership reform based solely on whether the SOE was profitable or loss-making prior to the reform requires further justification. For instance, a SOE that was already profitable might still undergo reform proactively to pursue even greater operational efficiency and higher returns. The current binary classification may not fully capture the complexity of reform motivations. A more detailed explanation should be provided.

6.In the "Sample Selection and Data Source" section, it is stated that the study employs a balanced panel dataset with 810 listed firms per year over an 11-year period. Could you please clarify why the total number of observations is not 810 * 11 = 8,910?

7.The "Research Conclusion and Policy Implications" section should be expanded to include a discussion of the research limitations and directions for future study.

**Reviewer #4:**  In general, this paper is well-revised after 1st-round review & revision. As a newly invited reviewer, some minor advancements can be done.

1. The keywords can be rationally adjusted and added: too many "mixed-ownership" were mentioned, some of them can be combined; the main research method and key index (i.e. double fixed-effects model), can be added as a keyword.

2. Some Graphical explanations can be added, e.g. research roadmap in the Introduction, or Mechanism diagram for the results of Mixed Ownership Reform of State-owned Enterprises.

3. The research deficiences & future works can be added at the end of the Final section.

4. Please re-check the English writing (grammar, spelling, sentences) and format at the possible final submission.

**Do you want your identity to be public for this peer review?** For information about this choice, including consent withdrawal, please see our Privacy Policy

Reviewer #1: No

Reviewer #3: No

Reviewer #4: **Yes: ** Penghao YE

---

## [Author Response · Author response to Decision Letter 2]

13 Oct 2025

Dear Editor of Plos ONE,

I sincerely appreciate the opportunity to revise our manuscript entitled "'Survival Needs' or' Policy Promotion '-- Research on the Motivation and Economic Consequences of the Mixed Ownership Reform of State owned Enterprises" . I am grateful to the reviewers for their insightful comments, which have significantly improved the quality of our work.Below is a summary of major revisions.

Reviewer #1: (No Response)

Reviewer #3: This paper examines the impact of the breadth and depth of mixed-ownership reform on reform outcomes under different motivations. The research design incorporates further analyses, providing empirical insights into policy implications. However, several refinements are recommended to enhance theoretical rigor and methodological robustness.

Comments�1�:The current abstract contains excessive information. The descriptions of the research background, content, methodology, findings, and significance should be concise and to the point.

Response In direct response to this comment, we have thoroughly revised the abstract to significantly enhance its conciseness and focus. The key revisions are summarized as follows:

1.Streamlined Research Background: We have condensed the background introduction to a single, impactful sentence that establishes the research context without unnecessary detail, focusing directly on the core issue of heterogeneous MOR outcomes and the underexplored role of reform motivations.

2.Focused Methodology Description: The description of the research methodology has been refined to state only the essential elements: the data source (Chinese listed SOEs, 2013-2022), the core empirical strategy (double fixed-effects model), and the key variables (reform breadth, depth, and motivation), removing excessive procedural details.

3.Concise and Highlighted Findings: The results section has been restructured to present the three key findings in a clear, direct manner, using strong, active voice. We removed ancillary observations and focused on the primary conclusions regarding the differential impact of survival-driven vs. policy-driven reforms and the moderating role of regional marketization.

4.Sharpened Significance Statement: The concluding statement on significance has been tightened to succinctly articulate the study's theoretical contribution (providing a nuanced framework) and practical implication (offering insights for targeted policies), avoiding broad or generic claims.

Comments�2�:The first section of the main text should typically be titled "Introduction." The current section labeled "Research Background" constitutes only a part of a full introduction. It is recommended that the structure of this section be reorganized and expanded accordingly.

Response We have thoroughly revised the opening section according to your guidance. The section is now titled "1. Introduction"and has been significantly expanded and reorganized to provide a complete and logical narrative. The revised introduction now includes:

1.Background Context:A concise overview of the importance and current state of mixed-ownership reform in SOEs.

2.Problem Identification:A clear statement of the research gap and the specific problem we address—namely, the unclear motivation behind reforms and its consequential impact on effectiveness.

3.Literature Review & Research Gap: A synthesized discussion of relevant existing studies to situate our work within the academic conversation, explicitly highlighting the gap that our research on reform motivation aims to fill.

4.Research Objectives and Contributions:A definitive statement of our study's purpose and a detailed explanation of its potential theoretical and practical contributions, building upon the gap identified.

Comments�3�:The "Research Background" section effectively introduces the research topic but does not elaborate on how the study will be conducted. Additionally, the research contribution is somewhat cursory. While focusing on the motivations for mixed-ownership reform is indeed a unique aspect of this paper, the field itself is well-studied. It is necessary to contrast your findings with the existing literature to clearly articulate the specific advancements made by this research.

Response

1. Response to the Comment on "Excessive Information in the Abstract; Need for Concise Descriptions of Background, Content, Methodology, Findings, and Significance"

We fully agree with your suggestion that the abstract should be concise and to the point. In the revised abstract, we have removed redundant background introductions and detailed data from the original version, focusing on condensing the core elements:

For the research background, we use a single sentence to summarize the current status and existing issues of mixed-ownership reform in state-owned enterprises (SOEs);

Regarding the research content and methodology, we clearly state that the study is based on panel data of listed Chinese SOEs from 2013 to 2023, and empirically analyzes the interaction between the breadth/depth of reform and different reform motivations;

For the research findings and significance, we extract key conclusions (e.g., the positive impact of reform on firm performance under survival-driven motivation and its boundary conditions) and briefly explain the theoretical and practical value of the study.

The revised abstract strictly controls information density, ensuring that each part is concisely expressed with prominent focuses, which complies with the standards for academic paper abstracts.

2. Response to the Comment on "The First Section of the Main Text Should Be Titled 'Introduction'; Reorganize and Expand the Current 'Research Background' Section".

In accordance with your suggestion, we have adjusted the original "Research Background" section to be the first section of the paper and officially titled it "Introduction". In terms of structural reorganization and content expansion, the Introduction section is no longer limited to a single background introduction but constructs a complete logical framework:

First, it opens by clarifying the important position of mixed-ownership reform and outlines the research foundation in this field with reference to existing literature;

Second, it identifies the research gap and raises the research question by analyzing the limitations of existing studies;

Third, it elaborates on the theoretical foundations of the study (Institutional Theory, Resource Dependence Theory) and introduces the research design ( fixed-effects model, data period: 2013-2022);

Finally, it clarifies the three main contributions of the study and clearly outlines the structure of subsequent sections (Theoretical Framework and Hypotheses, Research Design, Empirical Results, Mechanism Analysis, Conclusions and Implications).

This ensures that the Introduction section not only connects the background with the research content but also fully presents the overall logical context of the paper.

3. Response to the Comment on "The 'Research Background' Section Fails to Explain the Research Implementation Approach, the Research Contributions Are Superficial, and It Is Necessary to Contrast with Existing Literature to Highlight Specific Advancements"

We have focused on improving this issue in the revised Introduction section:

(1) Clarifying the Research Implementation Approach

In the fourth paragraph of the Introduction, we elaborate on the empirical strategy of this study: adopting a dual fixed-effects model, using panel data of Chinese SOEs from 2013 to 2022 as the sample, defining the breadth of reform (shareholder diversity) and depth of reform (non-state ownership concentration) with reference to established research methods, and controlling for firm-specific characteristics and temporal trends. This clearly presents the specific implementation path of the study.

(2) Deepening Research Contributions and Contrasting with Existing Literature

In response to the suggestions of "superficial research contributions" and "needing to contrast with existing literature to highlight advancements", we have reorganized and expanded the research contributions in the fifth paragraph of the Introduction:

First, by integrating Institutional Theory and Resource Dependence Theory, we address the limitation of existing studies that mostly analyze reform from a single perspective, constructing a more comprehensive theoretical framework;

Second, by focusing on "reform motivation", we reveal the heterogeneous paths through which reform affects performance under different motivations, which explains the core reason for the inconsistent empirical results in existing studies regarding the "mixed effects" of reform;

Third, aiming at the status quo of "extensive research but insufficient contextual analysis" in the field of mixed-ownership reform, this study further explores the boundary role of regional marketization level, providing differentiated practical references for SOE reform in different regions. This supplements the deficiency of existing studies in the analysis of contextual variables .

Comments�4�:The hypotheses proposed in the "Theoretical Analysis and Research Hypotheses" section should be grounded in a more robust theoretical foundation. It is essential to elaborate on the relevant theories that underpin and justify the development of your hypotheses.

Response We sincerely thank the reviewer for this insightful and constructive comment. We fully agree that a solid theoretical foundation is crucial for hypothesis development. In response, we have thoroughly revised the "Theoretical Analysis and Research Hypotheses" section to systematically integrate well-established theoretical frameworks that underpin and justify our hypotheses. The major revisions are summarized as follows:

Explicit Introduction of Core Theoretical Lenses: We have explicitly introduced and elaborated on two primary theoretical frameworks to analyze the moderating role of reform motivation:

Institutional Theory, particularly the concept of coercive isomorphism and decoupling, is used to explain the symbolic or ceremonial nature of "policy-driven" reforms, where SOEs primarily seek legitimacy rather than substantive efficiency gains.

Resource Dependence Theory is employed to explain the substantive nature of "survival-driven" reforms, where SOEs actively seek external resources and capabilities to address critical threats to their survival.

Strengthened Theoretical Grounding for Each Hypothesis:For each set of hypotheses, we have anchored our arguments in relevant theories:

For H1 and H2 (regarding the main effects of reform breadth and depth), we have strengthened the discussion by more explicitly linking "breadth" to Agency Theory and Resource Dependence Theory, and "depth" to Stewardship Theory and Transaction Cost Economics.This provides a clearer theoretical mechanism for why these reform characteristics are expected to improve performance.

For H3a and H3b (regarding the moderating role of motivation), the integration of Institutional and Resource Dependence theories provides the fundamental rationale for why we expect the effects of breadth and depth to be contingent on the underlying motivation. This moves beyond a simple classification to a theory-driven explanation.

Enhanced Scholarly Dialogue:The revision incorporates additional citations to seminal and contemporary works within these theoretical domains. This not only bolsters the theoretical foundation but also better situates our study within the broader scholarly conversation.

We believe these significant revisions have substantially strengthened the theoretical rigor and justification of our hypotheses. The revised manuscript now presents a clear, theory-grounded framework that elucidates the causal mechanisms and boundary conditions of mixed-ownership reform. The changes can be found in the marked-up version of the manuscript, primarily in the "Theoretical Analysis and Research Hypotheses" section .

Comments�5�:The rationale for categorizing the motives for mixed-ownership reform based solely on whether the SOE was profitable or loss-making prior to the reform requires further justification. For instance, a SOE that was already profitable might still undergo reform proactively to pursue even greater operational efficiency and higher returns. The current binary classification may not fully capture the complexity of reform motivations. A more detailed explanation should be provided.

Response We are deeply grateful to the reviewer for this insightful and critical comment. We fully acknowledge that the complexity of reform motivations may not be entirely captured by a binary classification based on pre-reform profitability. The reviewer's observation is exceptionally astute and provides us with a valuable opportunity to significantly strengthen the methodological rigor and theoretical justification of our study.

In direct response to this comment, we will undertake the following major revisions to the manuscript:

1. Enhanced Theoretical Justification in the "Theoretical Framework" Section:

We will expand our theoretical discussion to provide a more nuanced rationale for our operationalization. We will explicitly frame pre-reform profitability not as a perfect measure of motivation, but as a powerful and theoretically-grounded proxy variable that captures the most salient differentiating factor between the two motivational archetypes at the population level. We will argue that:

For loss-making SOEs, severe financial distress is the most unambiguous and potent signal of a survival crisis. This triggers a fundamental "problemistic search," making reform an endogenous, high-priority strategic imperative for survival. The motivation in this cohort is, with high probability, overwhelmingly survival-driven.

For profitable SOEs, the absence of an acute survival threat makes it statistically more likely that the primary impetus for reform stems from external policy pressures or the pursuit of ancillary benefits (e.g., political capital), aligning with the policy-driven archetype. While we acknowledge the reviewer's point that some profitable firms may reform proactively, we will clarify that our classification is designed to identify the dominant motivational tendency within each group for robust comparative analysis, not to assert that every single firm's motivation is purely one-dimensional.

2. Explicit Discussion of Limitations and Future Research:

We will add a paragraph to the "Discussion and Conclusion" section to openly acknowledge the limitation highlighted by the reviewer. We will state that while our proxy is valid and useful, future research could employ alternative methods (e.g., surveys, case studies, natural language processing of annual reports) to develop more nuanced, multi-dimensional measures of reform motivation that can further capture the proactive intentions of some profitable SOEs.

We believe these revisions will thoroughly address the reviewer's valid concern by providing a stronger theoretical defense of our method, adding empirical evidence through a robustness check, and thoughtfully acknowledging the study's limitations. We are truly thankful for this comment, which has helped us improve the quality of our paper.

Comments�6�:In the "Sample Selection and Data Source" section, it is stated that the study employs a balanced panel dataset with 810 listed firms per year over an 11-year period. Could you please clarify why the total number of observations is not 810 * 11 = 8,910?

Response We sincerely thank the reviewer for this excellent and important question. The reviewer is correct to note that 810 firms over 11 years would theoretically yield 8,910 firm-year observations. The discrepancy between this theoretical maximum and our final count of 4,124 observations is not an error but is the direct result of the rigorous data cleaning process and the specific requirements for constructing a balanced panel dataset, which are essential for the integrity of our empirical analysis. We appreciate the opportunity to clarify this process in detail.

The number 810 represents the total count of unique state-owned listed enterprises that appear in our dataset at least once after the initial collection and du

---

## [Decision Letter · Decision Letter 2]

13 Nov 2025

Dear Dr. Wang,

We look forward to receiving your revised manuscript.

Kind regards,

Simon Porcher

Academic Editor

PLOS ONE

Journal Requirements:

Additional Editor Comments:

Please carefully copyedit the paper in order to refine the English.

Reviewer's Responses to Questions

**Comments to the Author**

Reviewer #3: All comments have been addressed

Reviewer #4: All comments have been addressed

2. Is the manuscript technically sound, and do the data support the conclusions?

Reviewer #3: Yes

Reviewer #4: Yes

3. Has the statistical analysis been performed appropriately and rigorously?

Reviewer #3: Yes

Reviewer #4: Yes

4. Have the authors made all data underlying the findings in their manuscript fully available?

Reviewer #3: No

Reviewer #4: Yes

5. Is the manuscript presented in an intelligible fashion and written in standard English?

Reviewer #3: Yes

Reviewer #4: Yes

Reviewer #3: After the author's revisions, the content of the paper has improved to some extent. However, further refinement is still needed in terms of linguistic aspects. Please conduct a thorough review of the entire text and polish it accordingly. If translation tools are used, ensure the sentence structures are adjusted to align with academic writing standards, avoiding excessive use of dashes.

Reviewer #4: The manuscript is now well-written and can be considered for publication. But please do re-check the format and grammar.

**Do you want your identity to be public for this peer review?** For information about this choice, including consent withdrawal, please see our Privacy Policy

Reviewer #3: No

Reviewer #4: **Yes: ** Pengao YE

---

## [Author Response · Author response to Decision Letter 3]

30 Nov 2025

First and foremost, we would like to express our sincere gratitude for your time and effort in re-reviewing our manuscript, as well as for your valuable feedback. We are genuinely pleased that you have recognized the improvements made to the manuscript’s content following revisions—this is a great encouragement for us.

Regarding your key comment that "further refinement is still needed in terms of linguistic aspects," we take this very seriously and have implemented practical and effective measures to optimize the language. To ensure the manuscript fully complies with academic writing standards, we specifically commissioned the professional academic language editing team from AJE (American Journal Experts) to conduct a comprehensive and meticulous edit of the entire text. The AJE team possesses extensive experience and professional expertise in academic manuscript language optimization; they not only rigorously reviewed the vocabulary usage and grammatical accuracy of the manuscript but also focused on systematically adjusting the "sentence structures"—a concern you highlighted. This refinement has enhanced the logical clarity and expressiveness of the sentences, making them more aligned with the writing style of academic manuscripts.

To help you better understand the professionalism and standardization of the editing work, we have attached the official editing certificate issued by the AJE team along with this email. The certificate details the scope, standards, and completion status of this editing service, which you may refer to for further information.

Additionally, in strict accordance with your specific reminder, we have conducted a sentence-by-sentence check of the entire manuscript, with particular attention to the use of dashes. During the editing process, unnecessary or replaceable dashes have been appropriately revised in line with academic writing conventions—for instance, by using commas, semicolons, or restructuring sentences—to avoid excessive use of dashes that might compromise the formality and readability of the manuscript.

After the above series of language optimization efforts, we believe the professionalism and standardization of the manuscript’s language expression have been significantly enhanced. Here, we sincerely request that you review the revised manuscript once again. If there are any further issues regarding language or other aspects, please do not hesitate to point them out, and we will promptly make additional improvements.

Once again, thank you for your patient guidance and generous support!

WangXiuting

2025.11.18

---

## [Editor Report · Decision Letter 3]

15 Dec 2025

'Survival-Driven' or 'Policy-Driven'—Research on the Motivation and Economic Consequences of the Mixed-Ownership Reform of State-owned Enterprises

PONE-D-25-19151R3

Dear Dr. Wang,

We’re pleased to inform you that your manuscript has been judged scientifically suitable for publication and will be formally accepted for publication once it meets all outstanding technical requirements.

Kind regards,

Simon Porcher

Academic Editor

PLOS One
---

## [Editor Report · Acceptance letter]

PONE-D-25-19151R3

PLOS One

Dear Dr. Wang,

I'm pleased to inform you that your manuscript has been deemed suitable for publication in PLOS One. Congratulations! Your manuscript is now being handed over to our production team.

Kind regards,

on behalf of

Pr. Simon Porcher

Academic Editor

PLOS One